# Involvement of Extracellular Vesicles in Vascular-Related Functions in Cancer Progression and Metastasis

**DOI:** 10.3390/ijms20102584

**Published:** 2019-05-26

**Authors:** Shinsuke Kikuchi, Yusuke Yoshioka, Marta Prieto-Vila, Takahiro Ochiya

**Affiliations:** 1Division of Molecular and Cellular Medicine, National Cancer Center Research Institute, Tokyo 104-0045, Japan; kikuchi@asahikawa-med.ac.jp (S.K.); yyoshiok@tokyo-med.ac.jp (Y.Y.); mprietov@ncc.go.jp (M.P.-V.); 2Department of Vascular Surgery, Asahikawa Medical University, Asahikawa 078-8510, Japan; 3Division of Molecular and Cellular Medicine, Institute of Medical Science Tokyo Medical University, Tokyo 160-0023, Japan

**Keywords:** cancer progression, metastasis, extracellular vesicle, exosome, microRNA, angiogenesis, extracellular matrix, endothelial cell, platelet

## Abstract

The primary cause of mortality among patients with cancer is the progression of the tumor, better known as cancer invasion and metastasis. Cancer progression involves a series of biologically important steps in which the cross-talk between cancer cells and the cells in the surrounding environment is positioned as an important issue. Notably, angiogenesis is a key tumorigenic phenomenon for cancer progression. Cancer-related extracellular vesicles (EVs) commonly contribute to the modulation of a microenvironment favorable to cancer cells through their function of cell-to-cell communication. Vascular-related cells such as endothelial cells (ECs) and platelets activated by cancer cells and cancer-derived EVs develop procoagulant and proinflammatory statuses, which help excite the tumor environment, and play major roles in tumor progression, including in tumor extravasation, tumor cell microthrombi formation, platelet aggregation, and metastasis. In particular, cancer-derived EVs influence ECs, which then play multiple roles such as contributing to tumor angiogenesis, loss of endothelial vascular barrier by binding to ECs, and the subsequent endothelial-to-mesenchymal transition, i.e., extracellular matrix remodeling. Thus, cell-to-cell communication between cancer cells and ECs via EVs may be an important target for controlling cancer progression. This review describes the current knowledge regarding the involvement of EVs, especially exosomes derived from cancer cells, in EC-related cancer progression.

## 1. Involvement of Endothelial Cells in Cancer Progression

The primary cause of mortality among patients with cancer is tumor progression, better known as cancer invasion and metastasis [1]. When considering cancer progression, angiogenesis is a key tumorigenic phenomenon. It is a process in which a primitive vascular network grows and is remodeled into a complex network, ultimately developing into a complex mature vascular system. Understanding how cancer cells manipulate surrounding cells will lead us to further interesting research ideas. Endothelial cells (ECs) have been shown to be significantly involved in tumor progression, especially regarding tumor angiogenesis and tumor extravasation [2].

Angiogenesis involves EC activation, proliferation and migration. Tumor angiogenesis includes several steps: 1) injury of the basement membrane by cancer cells; 2) tissue destruction and hypoxia; 3) the activation of ECs; and 4) the presence of angiogenic factors, including growth factors and microRNAs (miRNAs). In the adult body, blood vessels remain in a quiescent state; however, various growth factors and miRNAs released from cancer cells play a major role in gene regulation, resulting in cell growth activation. Growth factors include vascular endothelial growth factor (VEGF), basic fibroblast growth factor (FGF), angiotensin, and transforming growth factor. In EC activation, significant associations have been found between tumor angiogenesis and miRNAs, including miR-126, miR-221/222, miR-23, and the miR-17-92 cluster [3,4,5,6,7,8,9,10,11]. The involvement of ECs via these growth factors and miRNAs derived from cancer cells is integral to tumor progression.

Tumor extravasation comprises multiple steps in which circulating tumor cells attach to ECs via major adhesion molecules, selectins, and integrins and then transmigrate through junctional gaps in endothelial monolayer, a process known as diapedesis [12]. The migrated tumor cells actively contract the ECs to open a junction gap with generating strong stress that push into the matrix and expressing proinflammatory signals [13]. Many vascular-related cells are commanded by cancer cells, similar to a conductor of an orchestra. Microthrombi and platelet aggregation are also characteristic of cancer and help excite the tumor environment by increasing the development of inflammation, leading to the activation of microvascular ECs and the formation of a metastatic microenvironment [14].

Cancer progression involves a series of biologically important steps in which cross-talk between cancer cells and cells in the surrounding environment is positioned as an important issue [15]. Various cell types share space in the cancer environment, and non-cancer cells are directly and indirectly affected by cancer cells via physical communication and the secretion of cytokines. Such actions activate non-cancer cells and change their phenotype to one similar to cancer cells. Thus, cell-to-cell communication is potentially an important target for controlling cancer progression. Over the last 10 years, our understanding of the role of extracellular vesicles (EVs) in cell communication has greatly increased. In this review, we focus on EVs including exosomes and microvesicles, which are responsible for interactions between vascular-related cells such as ECs and platelets and cancer cells in tumor angiogenesis and extravasation, which are the bases of cancer progression.

## 2. EVs in Cell-to-Cell Communication and the Roles in Cancer Research

EVs were initially thought to be nothing more than dust particles excreted by cells. However, it is now known that EVs carry various types of information derived from parental cells [16]. EVs have been studied worldwide and various fields recognize their role in cell-to-cell communication. They are involved in multiple biological responses including immunoresponses and inflammatory reactions, which were previously thought to be mainly regulated by chemokines and cytokines [17,18,19]. EVs are cell-derived membrane-bound vesicles, which consist of a lipid bilayer membrane and a small organella-free cytosol [20]. Circulating EVs comprise three types typically classified by their size and production process: exosomes, microvesicles, and apoptotic bodies [21]. In cancer-related EVs, exosomes and microvesicles commonly contribute to the modulation of a microenvironment favorable to cancer cells. Exosomes are EVs that are approximately 100 nm in size, which is smaller than microvesicles. There is some confusion between exosomes and microvesicles regarding a single cell type release, despite the production process; exosomes originating from multivesicular endosomes (MVE) lead to exosome secretion, whereas microvesicles are shed by the plasma membrane [22]. Although this review focuses on these two EV types, the accumulated data suggest that exosomes are more biologically important in cancer progression and metastasis.

All cell types can release EVs as avatars, which take over the functions of parental cells and can directly influence surrounding cells and distal targets by traveling in the extracellular space and circulation [23]. Thus, EVs have a focal function as well as a systemically important function. We can improve our knowledge of biological process by better understanding the roles of EVs. After being secreted by parent cells, EVs are incorporated by recipient cells via endocytosis [24,25]. Although routes of the recipient cell uptake remain elusive, these processes are acted through fusion of the EV membrane with the endosomal membrane and other mechanisms, such as nuclear transport of EVs or scanning of the endoplasmic reticulum [26,27]. Their lipid membranes are then digested, allowing the contents of EVs to be released and influence the recipient cells. The contents of EVs include various nucleic acids, lipids, and proteins. It has been shown that a substantial amount of RNA, especially miRNA, is found in <10% of EVs. Notably, miRNA can repress the activity of specific target mRNAs in intercellular communication [28]. Moreover, EVs are highly stable and can be transported in circulation without being degraded, thereby enabling cell-to-cell communication in the surrounding environment as well as at distant locations [25].

In recent years, cell-to-cell communication via EVs has attracted substantial attention in the field of cancer research. The accumulated findings have led to deeper understanding of the mechanisms of cancer progression and treatment. In particular, EVs have been studied in relation to tumor progression and metastasis, which directly relate to patient mortality. Cancer-related EVs are released from cancer cells as well as other cells such as ECs and platelets, which are involved in focal tumor progression and systemic circulation in cancer metastasis. Thus, it is very important to understand how cancer cells influence surrounding cells and distant cells via EVs in the tumor microenvironment. For example, the proinflammatory status and hypoxic conditions created by cancer cells modulate the phenotype of surrounding cells and promote the release of EVs from affected cells into the tumor microenvironment (Figure 1A–C) [29,30,31].

## 3. Function of Exosomes and Microvesicles in Cancer Progression

Like exosomes, microvesicles are circulating biopackages released from the plasma membranes of many types of cells, and they are functionally very similar to exosomes [32], also playing essential roles as conveyers of cell-to-cell communication in circulation [33,34,35]. Microvesicles play roles in inflammation, coagulation, and stromal cell activation related to cancer, and they are strongly involved in cancer progression and metastasis as well as thrombosis formation [36,37,38]. Due to the difficulty in isolating microvesicles from exosomes, little is known about the specific functional differences. However, some general points of difference have been reported [39]. First, microvesicles are larger than exosomes, with diameters of 100–1000 nm [40]. Conseza et al. showed differences in immune functions between exosomes and microvesicles derived from same mesenchymal stem cell condition medium in inflammatory arteritis [41]. In an in vivo study, exosomes provided a greater immunosuppressive effect than microvesicles, although the effects between the two were similar in vitro. Smaller nanoparticles like exosomes are less prone to opsonization, and exosomes and microvesicles have preferred locations that differentially affect their biodistribution and clearance [42]. Second, microvesicles are released from plasma membranes whereas exosomes originate from MVEs [43]. Although the membranes of both EV types are formed as a phospholipid bilayer, there is a clear difference in protein expression on their surfaces. The membranes of exosome are enriched in expression of various proteins that commonly include tetraspanins (such as CD9, CD63, and CD81), endosomal-sorting complexes required for transport (TSG101), and heat shock proteins (HSP60, HSP 70, and HSP90) [44,45]. By contrast, for microvesicles, integrins, selectins, and CD40 are representative surface markers [46]. The individual proteins expressed in exosomes are specific to the donor cell type [47], and can thus be used to help identify the origin of the exosomes and disease condition [48]. Such characterization has been widely used to investigate cancer biology and may have future clinical applications. For example, CD147 is highly expressed on the surface of exosomes derived from colorectal cancer cells and has been proposed as a potential diagnostic target for this type of cancer [49]. Furthermore, 80% of circulating microvesicles isolated from fresh blood samples were found to originate from circulating cells like platelets and ECs [50]. Platelet-derived microvesicles are known to contribute to tumor growth and metastasis [51]. Specifically, they directly enhance tumor growth through the release of potent growth factors that make the physiological state favorable to tumor growth, tethering, and dissemination [52]. Increases of platelet-derived microvesicles are strongly associated with the severity of gastric cancer [33]. In addition, elevated levels of microvesicles in blood and formation of venous thromboembolism (VTE) have been reported in patients with cancer [32], which are associated with advanced cancer and poor prognosis [50,53]. Cancer-derived exosomes are also related to increased VTE incidence due to high expression of tissue factor-coagulation factor complexes [54]. Muhsin-Sharafaldine et al. reported distinct phenotypic and functional differences between vesicle types in melanoma, showing that microvesicles have greater potential than exosomes to contribute to prothrombotic states and anti-cancer immunity [36]. As we thought, microvesicles have more specific function related to procoagulant activity in cancer progression compared with exosomes (Figure 1A).

Microvesicles can be released from ECs, neutrophils, monocytes, and lymphocytes [32]. EC-derived microvesicles can be pro- or antiangiogenic, depending on the stimuli causing their production [55]. They can regulate tumor angiogenesis, with predominantly proangiogenic effects, through the transportation of growth factors such as VEGF-A and FGF2 [56,57]. EC-derived microvesicles promote endothelial growth through direct interaction with ECs or via the activation of endothelial progenitor cells (Figure 1B) [58,59]. In patients with breast cancer, decreased levels of EC-derived microvesicles have been shown to be associated with better overall survival and disease-free survival rates after chemotherapy [59]. There are few articles studied about EC-derived exosomes. They have characteristics which contain several receptors related to transportation of macromolecules across gap junction. This function is fascinating, however information regarding ECs derived exosomes in the cancer field is still limited [60,61,62]. When considering the role of ECs in cancer progression, EVs derived from ECs seem to be still less significant compared with the change in phenotype and permeability of ECs by action of cancer cells. Therefore, we focus on involvement of cancer-derived EVs in vascular-related functions for understanding of the mechanisms of cancer progression in this review.

## 4. Contribution of EVs Derived from Cancer Cells to the Activation of ECs

Tumors are dependent on angiogenesis for their progression. EVs play an important role in transferring genetic information from cancer cells, including angiogenic miRNA, and mRNA, and even proteins to distant cells via the circulation [63]. Nazarenko et al. was the first to demonstrate that ECs were activated by tumor-derived exosomes through the VEGF-independent regulation of angiogenesis-related genes [64]. The activation by the exosomes enhances proliferation, migration, and sprouting of ECs by gene modulations, meaning that exosome-initiated EC regulation has a role in tumorigenic angiogenesis (Figure 1). They also showed that proteins contained by exosomes, such as CD106 and CD49d, were associated with the binding between cancer-derived exosomes and ECs, as well as their internalization. The exosome–EC interactions provide an interesting perspective for understanding the formation of the tumor microenvironment. As stated regarding the importance of exosomal miRNAs previously, it is a noteworthy fact that incorporation of exosomal miRNAs by ECs results in regulation of angiogenic-related genes and subsequent increase in VEGF and activation of the downstream signaling pathway [65,66,67,68]. Many exosomal miRNAs modulate EC phenotypes in this process [69,70].

Hypoxia is involved in cancer progression because it triggers the proangiogenic pathway that involves the cancer-derived exosomes secretion (Figure 1C) [71]. ECs are activated to a greater extent by exosomes derived from hypoxic cancer cells [68], which elicit a change in the proangiogenic EC phenotype through the upregulation of heparin-binding epidermal growth factor signaling and beta-catenin signaling, or uptake of exosomal miR-210 increased by hypoxia [71,72,73]. miR-210 is a representative miRNA associated with hypoxia and is highly expressed in exosomes derived from malignant cancer cells relative to non-malignant cells [74]. Exosomal miR-210 enhances tumor angiogenesis via uptake by ECs and suppression of specific target genes. Indeed, miR-210 is listed as a prognostic factor in breast cancer [75]. It is well established that hypoxia increases the release of exosomes from cancer cells. In this regard, Rab27a is a gene, which up-regulates exocytosis of MVEs, and has been researched as a key gene to increase of exosome secretion from cancer cells. [76].

The activation of ECs also affects tight junctions, which prevent leakage of transported solutes, passing of molecules, and cell invasion (Figure 2A). Exosomes from metastatic cancer cells modulate tight junctions via activation of ECs and promoted tumor progression [77]. In lung cancer, the loss of the tight junction caused by cancer-secreted exosomal miR-23a results in increased angiogenesis [78]. Hence, exosomes derived from cancer cells affect EC functions not only through VEGF-induced proangiogenic effects but also via other angiogenesis-related processes [79]. For instance, EVs contain antiapoptotic long non-coding RNA as well as miRNA preventing senescence, allowing the formation of blood vessels [80,81]. In these ways, cancer cells act on ECs to promote angiogenesis, resulting in tumor growth.

## 5. Modulation of Endothelial Permeability by Cancer Cells and the Role of Cancer-Derived Exosomes

The effect of EV-associated miRNAs on the vascular system is a key part of the mechanisms that underlie the spread of cancer cells, acting as a potential modulator of ECs [82]. Cancer-derived exosomes are incorporated into ECs, and their contents regulate target genes, including transcription factors that modulate vascular permeability [83]. Endothelial permeability is increased to a greater extent by exosomes from metastatic cancer cells rather than by those from non-metastatic cells (Figure 2A); this results from cytoskeletal-associated proteins, such as thrombin, which activate the RhoA/ROCK pathway [77]. In colorectal cancer, exosomal miRNAs (including miR-200c, miR-141 and miR-429) derived from cancer cells were associated with changes in adjacent endothelial barriers through the regulation of ZEB proteins [84,85]. Similarly, the exosome-mediated transfer of miR-23a and miR-105 targets the tight junction protein ZO-1, destroying the integrity of natural barriers against metastasis [78,86]. The loss of tight junctions by exosomal miRNAs derived from cancer cells has a negative influence on patient prognosis. In triple-negative breast cancer, which is related to poor survival, poor prognosis can be explained by high expression of miR-939 in MDA-MB-231 cell-derived exosomes. Exosomal miR-939 increases the permeability of the EC monolayer by downregulating VE-cadherin [87]. In contrast to cancer-derived exosomes, ECs protect blood barriers by secreting miRNA-145-5p and miR-200 and through the upregulation of connexin-43 expression in gap junctions [85,88].

The blood–brain barrier (BBB) is an exceptionally important clinical EC barrier against brain metastasis. The BBB restricts the penetration of molecules of various sizes into the brain, and its permeability is associated with metastasis [89]. An improved understanding of the function of exosomes has helped with the research of the mechanisms that underlie brain metastasis. Proteins packaged in EVs released from glioblastomas, including VEGFA and semaphorin 3A, contribute to endothelial permeability [90,91]. Similarly, exosomal miRNAs have been reported to affect the homeostasis of tight junctions of BBB. Tominaga et al. reported that miR-181c contained in exosomes from brain metastatic breast cancer cells can destroy the BBB. ECs incorporated the exosomes and miR-181c degraded *PDPK1*, a gene involved in actin dynamics, resulting in the promotion of extravasation of cancer cells and subsequent brain metastasis [92].

The permeability of ECs and extravasation of cancer cells through cancer-derived exosomes are essential phenomena in brain metastasis. There is an attempt to apply EVs as a therapeutic tool by using the characteristics of modulation of ECs to the patient’s advantage. Small interfering RNA (siRNA) holds great therapeutic promise regarding EV-dependent delivery; however, the delivery of siRNA across the BBB still remains an experimental issue. Yang et al. evaluated the potential for exosomes to be used to deliver drugs across the BBB and into brain, in which ECs-derived exosomes penetrated BBB compared with cancer-derived exosomes. CD63 expressed more on the surface of ECs-derived exosomes is likely to aid the BBB penetration, leading that ECs-derived exosomes were optimized more frequently than caner-derived exosomes in brain drug delivery [93]. As a result, they demonstrated that EC-derived exosomes could deliver siRNA across the BBB. They showed that siRNA delivered in a natural carrier, such as EC-derived exosome, could inhibit VEGF and reduce cancer cell activity [62].

## 6. Roles of EVs in Endothelial-to-Mesenchymal Transition

The development of the tumor environment requires ECM remodeling, angiogenesis, and the presence of cytokines and growth factors for rapid tumor growth, as previously described. This environment also comprises a heterogeneous group of cells, including stromal cells and cancer cells [94]. Non-cancer stromal cells are significantly affected by tumors. Mesenchymal stem cell, fibroblasts, pericytes, adipocytes, macrophages, and immune cells change their phenotypes after being activated by various tumor factors. Released soluble factors, miRNAs, and exosomes can change the phenotype of the surrounding cells to one that facilitates the tumor environment [95,96,97,98,99,100]. The phenotypes of ECs can vary according to the surrounding pathological environment, forming distinct differentiated cell types fitted with the local microenvironment [101]. The endothelial-to-mesenchymal transition (EndMT) is a phenotypic change in which ECs become cancer-associated ECs, losing EC markers and acquiring a mesenchymal function (Figure 2B) [102]. EndEMT is characterized by several biological changes: 1) the loss of endothelial markers and cell–cell junctions, along with cytoskeletal changes involving CD31, von Willebrand factor Vlll, and E- and VE-cadherin; 2) the acquisition of invasive migratory and proliferative properties; 3) the enhanced expression of the cell contact proteins N-cadherin, beta1-integrin, and fibronectin; 4) the gain of mesenchymal markers (FSP-1 and alfa-smooth muscle actin); and 5) upregulation of the transcription factors Snail and Slug [102,103,104,105]. This biological rendering corresponds to enhancement of contractile properties and a myofibroblast-like phenotype as a cancer-associated like ECs [106]. EndMT is necessary for metastatic transendothelial migration, a potential mechanism of metastatic extravasation known to be a complex phenomenon [107].

EVs from cancer cells act as propagation carriers, transferring exosomal genomic information to the ECs for EndEMT [108]. Cancer-derived exosomes are incorporated by ECs and stromal cells, inducing an inflammatory phenotype in the recipient cells [109]. Paggeti et al. demonstrated that exosomes derived from leukemia strongly influenced the functions of ECs, such as proliferation, remodeling of the actin cytoskeleton, migration, and angiogenesis through incorporation of the exosomes by ECs [110]. Using melanoma cells, Yeon et al. showed that the number of cancer-associated ECs after EndMT increased by cancer-derived exosomes in a dose-dependent manner, with the induction of morphological and molecular changes in in vitro [111]. Interestingly, they also reported that exosomes derived from mesenchymal stem cells suppressed EndMT and restored cells from a cancer-associated phenotype to their original status [111]. The involvement of cancer-derived exosomes in EndMT remains unclear; however, relevant data are gradually accumulating, shedding light on this mechanism. For instance, it has been reported that exosomal Annexin A1 and RAC1/PAK2 induce EndEMT, whereas miRNA-200 suppresses EndEMT [85,112,113].

## 7. Involvement of EVs in Cancer Progression through Matrix Remodeling and Endothelial Cell Activation

Cancer cells establish physical contact with surrounding structures, including blood vessels, nerve tissues, and smooth muscles [114]. Matrix remodeling is an essential part of cancer progression. It would be interesting to know how cancer-derived EVs are involved with matrix remodeling in terms of its key function being intercellular communication. In cancer progression, matrix metalloproteinases (MMPs), enzymes that degrade the extracellular matrix (ECM) play important roles in tissue homeostasis and cell invasion. Cancer cells secrete MMPs into their exosomes and help constructing the matrix [115]. Quadir et al. investigated the functional differences and molecular consequences of normal cells responding to exosomes derived from normal cells compared with those derived from cancer cells. They found that cancer exosomes, but not normal exosomes, modulated the expression of matrix remodeling related genes, which is associated with cancer-associated pathology via transcriptome reprogramming of recipient cells [116]. Those genes included *BBOX1*, which is essential for transport of fatty acids, and *EFEMP1*, which regulates cell morphology and motility. These genes are also associated with cancer development and tumor invasiveness [117,118]. Taken together, these findings show that cancer-derived exosomes play important roles in matrix remodeling.

Cancer cells can enhance angiogenic and metastatic potential with reduced cell-to-cell and cell-to-ECM adhesion (Figure 2C). This is achieved by the regulation of the expression of several genes, such as annexin A6 and the oncogenic protein tyrosine kinase Kit [119,120]. The contents of the exosomes are associated with cell adhesion molecules, the regulation of the cytoskeleton, cell-to-cell communication, ECM–receptor interactions, and focal adhesion [121]. Exosomes hold a firm position as a representative in cancer cell secretion and are greatly associated with ECM degradation and remodeling (Figure 2D) [122]. In particular, exosomal MMP-1 derived from cancer cells is a representative indicator of ECM degradation in tumor invasion and metastasis [120,123]. This is consistent with the report of Bobrie et al. who showed that the reduction of exosomal secretion by a blockade of Rab27a decreases the secretion of cytokines and MMPs, resulting in inhibition of tumor growth and metastasis [124]. Hood et al. investigated the involvement of metastasis of melanoma-derived exosomes in lymph nodes (a primary metastatic site of melanoma). They demonstrated that the exosomes approached sentinel lymph nodes easily and induced the expression of genes associated with cell recruitment, ECM remodeling, and the production of vascular growth factors [125].

Hypoxia induces not only more angiogenesis but also ECM remodeling via an increase in exosome secretion, resulting in cancer progression [76]. An increased ECM degradation via the modulation of MMP3, MMP9, and MMP13 and further angiogenesis in various cancer cell lines under hypoxia have been reported [121]. Similarly, heparanase, an enzyme that degrades ECM and is upregulated in more aggressive cancer cell lines, regulates exosome secretion and composition, leading to the spread of tumor cells and EC invasion [126].

Based on the above, EVs derived from cancer cells are substantial promoters of cancer progression via the remodeling matrix. These effects are greatly associated with tumor angiogenesis.

## 8. Vasculogenic Mimicry: ECs-unrelated Vasculature and Possible Involvement of EVs

There is increasing evidence regarding the mechanism for the heterogenous blood supply because the specific type of vasculature is associated with patient prognosis [127]. Recent findings indicate that the microvasculature associated with a tumor is completely different to that associated with normal tissue, and even independent EC vasculature has been found [128]. The connecting networks inside tissues, included ECM secreted by cancer cells and have a shape of a vessel surrounded by basement membranes [129,130,131]. The hollow channels lined up by cancer cells comprise red blood cells; however, immunohistochemistry findings have suggested that there was no EC in this area because no expression of Factor VIII-related antigen, CD31 or CD34 was found [132]. Such vessel networks, described as vasculogenic mimicry (VM), facilitate tumor perfusion independently from tumor angiogenesis (Figure 2E). A typical specific evaluation method is to stain the basement membrane of the tumor ECM with periodic acid–Schiff; where it has been shown that a positive pattern is significantly associated with the metastatic ability of the cancer. An important mediator of VM formation is hypoxia, which is also induced in normal angiogenesis involved with ECs since deletion of HIF-1α reduced VM formation [133,134].

This form of non-endothelial microcirculation has been recognized in various types of cancer tissues. The mitogens that form VM vary according to the cell line. For example, VM formation has been identified in hepatocellular carcinoma cells and VM has been induced by hepatocyte growth factor rather than by VEGF [135]. Epithelial malignancies associated with the Epstein–Barr virus develop VM formations that are dependent on VEGF. Various mitogens stimulate cancer cells, resulting in the increased expression of stemness genes such as Oct4 and Sox2 concomitant with the pluripotency of embryonic stem cells [135,136]. Wagenblast et al. investigated VM as a driver of metastasis using a mouse model of breast cancer heterogeneity, in which two genes (serpin peptidase inhibitor E2 and secretory leukocyte peptidase inhibitor) played an important role in the phenotypic change of tumor cells into endothelial-like cells [137]. Indeed, a member of the Serpin family is closely associated with cancer progression: SERPINA3 is overexpressed in gliomas, resulting in poor prognosis for patients with this condition [138]. As yet, there has been no report of the involvement of exosomes in VM; however, exosomes affect the molecular programs involved in matrix modulation and remodeling [116], which suggests that it is plausible that cancer-derived exosomes may be involved in VM formation.

## 9. Conclusions

ECs play fundamental roles in tumor progression. Tumorigenic angiogenesis is a key phenomenon that arises due to relationships between cancer cells and ECs, such as tumor extravasation, loss of tight junctions, ECM remodeling, and angiogenesis. Tumor metastasis, as well as focal tumor progression, is strongly associated with the activities of ECs and cancer cells. Understanding these phenomena will allow us to acquire useful knowledge and develop new insights in cancer research fields, under the assumption that EVs are available as a novel therapeutic agent in near future. During tumorigenic angiogenesis, EVs are derived from cancer cells, platelets, and ECs, which are a representative source of cell-to-cell communication. Microvesicles shed from platelets diffuse systemically in circulation and have procoagulant and inflammatory effects in patients with cancer. Exosomes related to cancer progression are mainly derived from cancer cells and contribute to EC-associated microenvironment activities. ECs activated by exosomes achieve a proliferative, migratory status and show modulation of their cytoskeletons. Cancer cells can progress focally and distally via their exosomes and manipulation of the surrounding cells, especially ECs. Thus, EVs have a significant influence on various aspects of cancer progression, including metastasis and the promotion of tumor angiogenesis. The basic and clinical importance of the relationship between vascular-related functions and the involvement of EVs has been corroborated by many evidences of cancer research. We believe that regulation of cell-to-cell communication between vascular-related cells and cancer cells by EVs may be potential therapeutic targets to inhibit tumor extravasation and subsequent cancer progression.

## Figures and Tables

**Figure 1 ijms-20-02584-f001:**
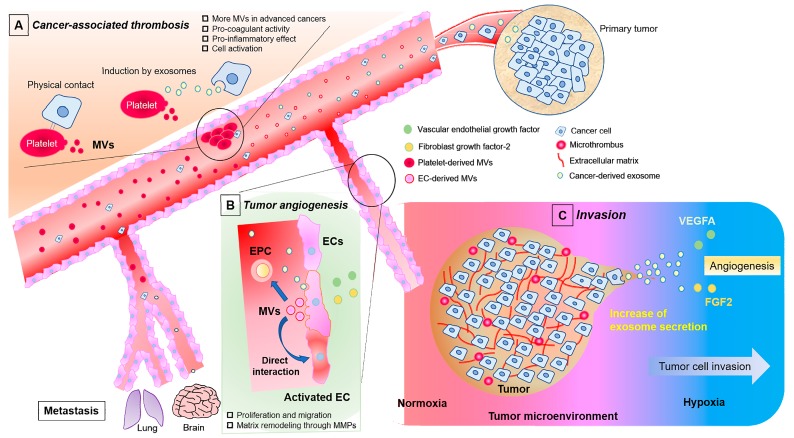
Involvement of ECs, platelets, and EVs in cancer progression, including angiogenesis, invasion, and metastasis. EVs play important roles in these processes through their function of cell-to-cell communication. Cancer cells develop favorable microenvironments by promoting angiogenesis and procoagulant and proinflammatory activities via the secretion of exosomes and hypoxia-induced growth factors. Platelets are activated via physical contact with cancer-derived exosomes, thereby resulting in increased secretion of platelet-derived microvesicles and the subsequent development of coagulation (**A**). ECs are activated by VEGF-A and FGF-2 released from cancer cells under hypoxic conditions and by circulating cancer-derived exosomes. Activated ECs shed microvesicles and activate other ECs via the microvesicles and direct interaction (**B**). Hypoxia is involved in tumorigenesis. A hypoxic microenvironment created by cancer cells triggers the proangiogenic pathway that involves the cancer-derived exosomes secretion (**C**). Circulating cancer cells and EVs from cancer cells, platelets, and ECs are strongly associated with cancer progression, including metastasis. EC, endothelial cell; EV, extracellular vesicle; VEGF, vascular endothelia growth factor; FGF, fibroblast growth factor; MV, microvesicle; EPC, endothelial progenitor cell.

**Figure 2 ijms-20-02584-f002:**
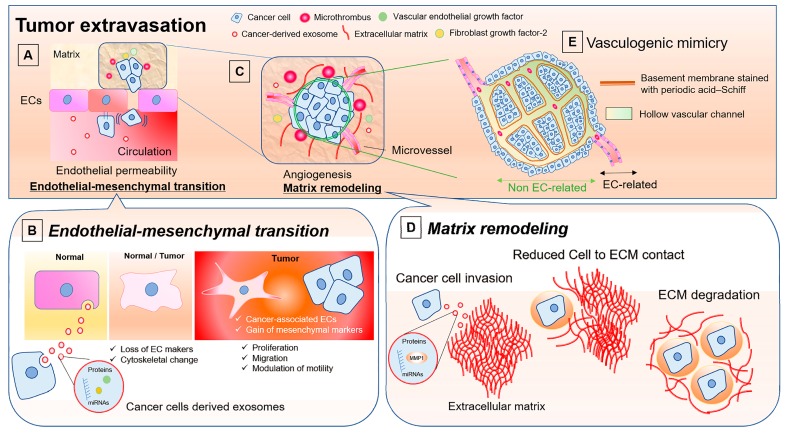
Effects of cancer-derived EVs on ECs in tumor extravasation. Attachment of circulating cancer cells to ECs and cancer-derived exosomes modulates tight junctions by activating ECs (**A**). Exosomes include proangiogenic miRNAs and growth factors and are involved in changing the phenotype of ECs. Activated ECs lose their markers and undergo a change in cytoskeleton. They gain mesenchymal markers and become cancer-associated ECs, which are proliferative and migratory. Endothelial-mesenchymal transition increases endothelial permeability and cancer cell invasion (**B**). The cancer cells induce inflammation by forming microthrombi and releasing growth factors, resulting in cancer cell growth and tumor angiogenesis (**C**). Cancer-derived exosomes are active in matrix remodeling via reduction of cell-to-ECM contact and degradation of the ECM through exosomal proteins, such as MMP-1 (**D**). When the cancer growth scaffold is established, the tumor microenvironment develops with a cancer-original vasculature that is not involved with ECs, known as vasculogenic mimicry (**E**). The vasculature may be regulated by cancer-derived exosomes, which are strongly associated with cancer progression. EC, endothelial cell; EV, extracellular vesicle; ECM, extracellular matrix.

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
