# Peer review of "Involvement of Extracellular Vesicles in Vascular-Related Functions in Cancer Progression and Metastasis"

_ijms, 2019, doi:10.3390/ijms20102584_

Round 1
Reviewer 1 Report
It is a well written and up to date review.
Author Response
We are greatful to the reviewer for this contribution.
Reviewer 2 Report
In this manuscript the Authors present the current state of knowledge on the vascular-related function of extracellular vesicles in cancer progression and metastasis.
In my opinion the article is logically arranged and grammatically well written. The Authors review carefully the available literature and provide a lot of the evidences for the proof of concept in the well-selected paragraphs. The references seem to be adequate.
Because studies on extracellular vesicles, especially those related to their functions and the possibilities of using them for diagnostic and therapeutic purposes, are becoming more and more common in the field of cancer research, I believe that this article will be interesting for many readers.
I recommend this manuscript for publication in its present form.
Author Response
We appreciate the check of this manuscript regardless of how you busy are. The comments of the reviewer encourage us to lead further contribution in this research field.
Reviewer 3 Report
This is a very informative and beautifully written review on a specific and interesting topic from authors that are expert in the relative field.
Only two suggestions:
A. Paragraph 2, line 89: The statement "Their lipid membranes are then digested, allowing the contents of EVs to be released and influence the recipient cells" may be misleading. After endosomal uptake, digestion of the membrane and pouring of the content into the cytoplasm has never been demonstrated. Rather, fusion of the EV membrane with the endosomal membrane and other mechanisms, like nuclear transport of EVs (Rappa et al., Oncotarget 2017; Santos et al., J. Biol. Chem. 2018) or scanning of the ER (Heusermann et al., J. Cell Biol. 2016) could be mentioned.
B. Paragraph 8 could be deleted or reduced because a role of EVs in vasculogenic mimicry has not yet been demonstrated.
Author Response
We appreciate your constructive comments. The following sentence is added in paragraph 2 of revised manuscript (Page 3, Lines 94-97)
Although routes of the recipient cell uptake remain elusive, these processes are acted through fusion of the EV membrane with the endosomal membrane and other mechanisms, such as nuclear transport of EVs or scanning of the endoplasmic reticulum [26, 27].
Regarding paragraph 8, we agree that vascular mimicry has not yet proven the involvement with EVs. However, this concept seems to be an important pathological status to understand further contribution to cancer progresson. Vascular mimicry is not well known among cancer-related researchers, and involvement of EVs should be interesting. We are greatful to get your understanding.
Reviewer 4 Report
This is a very nice review on a topic that I believe will be of interest to many readers. I really appreciate the style of the manuscript with no redundant text so that the reader gets much information within only several pages of the text. I have no requirements for the authors. I can recommend the manuscript for publication.
Author Response
Thank you for the reassuring comments. We hope that this manuscript helps lots of people all over the world, who reseach EVs and cancer.